# Prediction of Ground Surfaces by Using the Actual Tool Topography

**Rolf Hockauf \*, Volker Böß, Thilo Grove and Berend Denkena**

Institute of production engineering and machine tools, Leibniz Universität Hannover,
30823 Garbsen, Germany; Boess@ifw.uni-hannover.de (V.B.); Grove@ifw.uni-hannover.de (T.G.);
Denkena@ifw.uni-hannover.de (B.D.)
**\*** Correspondence: Hockauf@ifw.uni-hannover.de; Tel.: +49-(0)511-762-4788

**Abstract:** This paper presents a prediction model for ground surfaces that uses the actual grinding wheel topography to perform a grinding simulation. Precise knowledge of expected machined surfaces plays an important role in process planning. Here, the main criterion is the achievement of the components' function after manufacturing. Therefore, it is essential to consider the surface roughness to enable a function-orientated workpiece surface. The presented approach uses a real grinding tool topography, which is measured by a 3D laser triangulation sensor in the machine tool. After a data processing step, the measured topography is imported into a material removal simulation. A kinematic simulation of the realistic ground surface enables the data-based confirmation of the envelope profile theory for the first time.

**Keywords:** grinding; dexel-based kinematic simulation; surface prediction; grinding tool measurement; laser triangulation; envelope profile

---

## 1. Introduction

The state of the surface largely affects the properties of a component. It influences haptic, optics and tribological properties as well as the lifetime. For the construction of components and the subsequent manufacturing, these facts have to be considered. So far, it is not possible to predict the exact surface of a ground area [1]. In the past, a theory based on the envelope profile of a grinding tool was entrenched following the hypothesis that the highest point on the circumference of the grinding wheel generates the envelope profile and the workpiece surface [2]. In absence of knowledge about the exact tool topography, this theory has never been proven. A reason for this is that actual tool topography has not yet been considered in simulation models. A good summary of the simulation of grinding processes is given in the keynotes of Tönshoff and Brinksmeier [3,4]. They present many simulation models, which predict the process forces and temperature, chip thickness and subsurface properties as well as the surface topography. Most of those use simplified or modelled grains on the grinding tool surface. These tool models consist of octahedrons, tetrahedrons [5], balls [6–9], cones [9–11], ellipsoids [12–14] or a mixture of these [15,16]. The distribution of grains in surface and volume of the grinding tool is determined by random algorithms or modification of the actual tools. Due to these approaches, it is mostly possible to predict the roughness value of ground surfaces in a high accordance to measured values. Unfortunately, the roughness value cannot demonstrate the characteristic of a surface as Thomas has shown (Figure 1) [17].

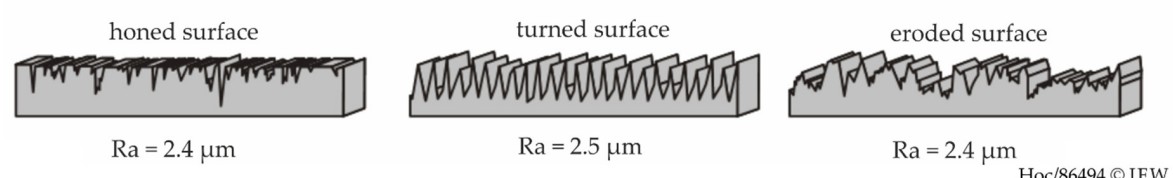

**Figure 1.** Different characteristics of surfaces with same roughness value [17].

Only Inasaki considers the actual tool topography in his investigation. Due to lack of technology at the time, he could only take a $2 \times 2$ mm$^2$ tool topography into account. With this approach, he predicted a three-dimensional workpiece for the first time [18]. A verification of the envelope profile theory was not possible with this approach.

However, the present investigation considers the whole topography of a grinding wheel. Therefore, it is possible to prove the envelope profile theory and predict the exact workpiece surface. Besides the roughness values, the correlation coefficient r between the related simulated and measured surfaces is considered to describe the different characteristic of the workpiece roughness (cf. Figure 1).

## 2. Experimental Setup

For an accurate simulation result, the import data for the tool model should be as realistic as possible. Therefore, three measuring devices were compared to each other and the laser triangulation sensor was chosen [19]. It provides the best method of measuring time and accuracy. The use of cerec optispray—a product for dental applications—unifies the reflection signal from the tool and enables measurement of the grinding tool with a 30 mm diameter and an 8 mm width in only 6.5 s. The point distance in the topography is 7 μm in circumference and 10 μm in width direction. As shown in Figure 2, the laser triangulation sensor is fixed to a holding plate, which is clamped in the machine vise of the machine tool.

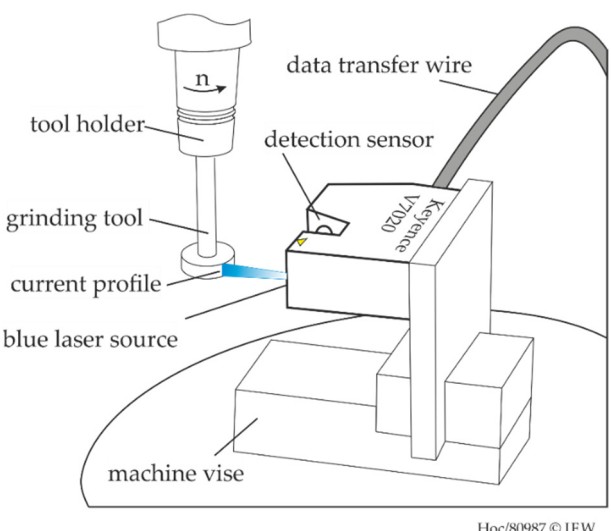

**Figure 2.** Setup of in-machine measuring system.

The result of the in-machine measurement is a topography with 13,330 profiles in peripheral direction, which consist of 800 points each. Despite spray-coating with a particle size of 1–5 μm, small artefacts occur in the raw data. A misfit filter in μ-Soft analysis was used to remove the artefacts and fill the missing areas with interpolated points.

For the grinding experiments, a Röders RFM 600 DS machine tool is used. It is a five-axis high-precision milling machine with rotary tilting table. The rectangular-shaped workpieces ($50 \times 50 \times 7.4$ mm$^3$) are made of SAE 1045 with a Rockwell hardness (cone) of 30 HRC. The workpiece

is clamped in the machine vise and ground with aqueous emulsion at the small flank so that nearly the whole tool width creates the workpiece surface. Due to this setup and a cutting depth of 50 μm, every stroke of the tool removes a volume $V_w$ of 18.5 mm³. After a defined number of strokes, the tool topography is measured to determine the wear. Additionally, the workpiece roughness profile is determined using a stylus instrument MarSurf LD130. Table 1 describes the measuring intervals.

**Table 1.** Measuring intervals during the wear investigation.

| Number of Strokes (á 50 μm) | Frequency of Measurement | Removed Volume between Each Measurement |
| --- | --- | --- |
| 1–5 | after each stroke | 18.5 mm³ |
| 5–15 | after two strokes | 37 mm³ |
| 15–60 | after 5 strokes | 92.5 mm³ |
| 60–285 | after 25 strokes | 462.5 mm³ |
| 285–2135 | after 50 strokes | 925 mm³ |

The described wear investigations were performed with a cutting speed of $v_c$ = 30 m/s, a feed speed of $v_f$ = 1000 mm/min with a galvanically bonded cubic boron nitride tool and a mean grain diameter of 151 μm. By measuring the whole topography of the grinding wheel, it is possible to describe the surface-generating tool parts. This way, it is possible to verify the envelope profile theory.

After the wear investigations, a material removal model is established in IFW CutS to predict the actual workpiece topography [20]. IFW CutS is a dexel-based kinematic material removal software, which has been developed at the IFW for the last 15 years. The novelty in the present investigation is that the measured tool topography with about 10.6 million points is used as tool representation without further simplification. The tool data is imported from an ASCII file, which consists of ordered height values from the laser triangulation sensor. These points are transformed into triangles to a closed surface according to the original cylindrical coordinate system as shown in Figure 3.

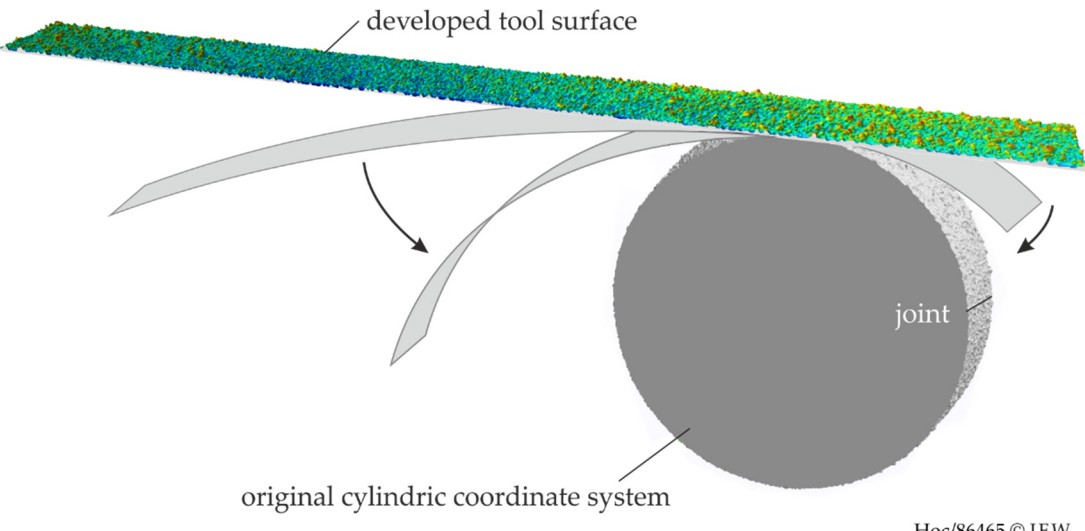

**Figure 3.** Retransformation of the tool topography in original cylindrical coordinate system.

The simulation moves the tool model by reproducing the kinematic movement using NC-code and calculating the trace of the relative motion between the tool and the workpiece. This trace—the so-called "sweep volume"—engages the workpiece. In the next step, the intersection is removed. Due to the high number of measured points, a high computation effort is necessary resulting in long simulation time. Thus, only 20 rotations were simulated, which correspond to about 1 mm in feed. However, the simulation of more rotations would generate an identical roughness profile in the simulation because two rotations would theoretically be sufficient to represent a stationary process. After the simulation of the workpiece surface, a comparison between simulation and experiment

was performed. The comparison uses the ratio of average surface roughness Rz and the correlation coefficient after Pearson [21], between the measured $x_i$ and simulated $y_i$ of the number of $n_r = 800$ workpiece profile points (Equation (1)).

$$r = \frac{\sum_{i=1}^{n_r} (x_i - \overline{x})(y_i - \overline{y})}{\sqrt{\sum_{i=1}^{n_r} (x_i - \overline{x})^2} \sqrt{\sum_{i=1}^{n_r} (y_i - \overline{y})^2}} \tag{1}$$

## 3. Results and Discussion

As explained in Section 2, the wear investigations were performed by measuring and grinding alternately. The laser triangulation method enables the measurement of exactly the same area in each iteration of the wear investigation. Figure 4 depicts the change of the grinding tool topography at different levels of removed workpiece volume $V_w$ during the whole investigation. It can be seen that the first grinding steps reduce the highest grain peaks and decrease the roughness of the whole topography. Between a removed workpiece volume of 1500 and 15,000 mm$^3$, the visualization of the topography appears in a nearly stable state. This contradicts the expectation that the wear of the tool topography is a constant process. This expectation is further supported by the observation that above 15,000 mm$^3$ until the end of the wear investigations at 37,600 mm$^3$, the topography roughness keeps decreasing.

Due to the unexpected wear behaviour, the observation was continued with tool surface roughness values. Especially the reduced peak height characterizes well the described observations, as Figure 5 demonstrates. During the first grinding strokes, the $S_{pk}$ value decreases from 44 to about 30 μm. Figure 4 shows this rapid wear too. A stable wear state, as shown in the middle of Figure 4, cannot be verified by the roughness graph, which confirms the expectation of constant wear. The decrease continues linearly until a reduced peak height of about 10 μm and the end of the tool's lifetime. If the clogging did not occur at the end of the tool life, the roughness values would approach asymptotically to a theoretically minimum.

Besides the wear examination, the surface-generating tool sections can now be described by using the envelope profile. Therefore, the highest points of each of the 800 measurements in tool width direction (laser profile length) and its position on the lateral tool surface are identified using Matlab. Figure 6 shows these points in dependency of the wear state. It should be noted that the number of surface-generating points is always 800 in these diagrams due to the length of the laser profile. The distribution of the highest points does not change dramatically during the wear investigations but the surface generating points stay concentrated in two sections of the tool's lateral surface. Possible reasons for this fact are a concentricity error of the tool and tool holder as well as a slightly ovality of the tool's basic body. In case of hand-guided galvanically bonded tools this should not be an issue. However, in an automated machining process, only a fraction of the whole tool surface is used, and the rest is not in contact. The form error of the investigated tools (besides the present results) with a mean grain diameter of 151 μm was measured in a size of between 40 and 90 μm. The laser triangulation measurement is hand-triggered, so that the involved surface sections in Figure 6 are not exact at the same position. At the end of the wear investigations, clogging was measured at the surface. Due to the decreased tool roughness, the chips cannot be removed completely so that clogging appears. It defines the limit of tool life for this investigation.

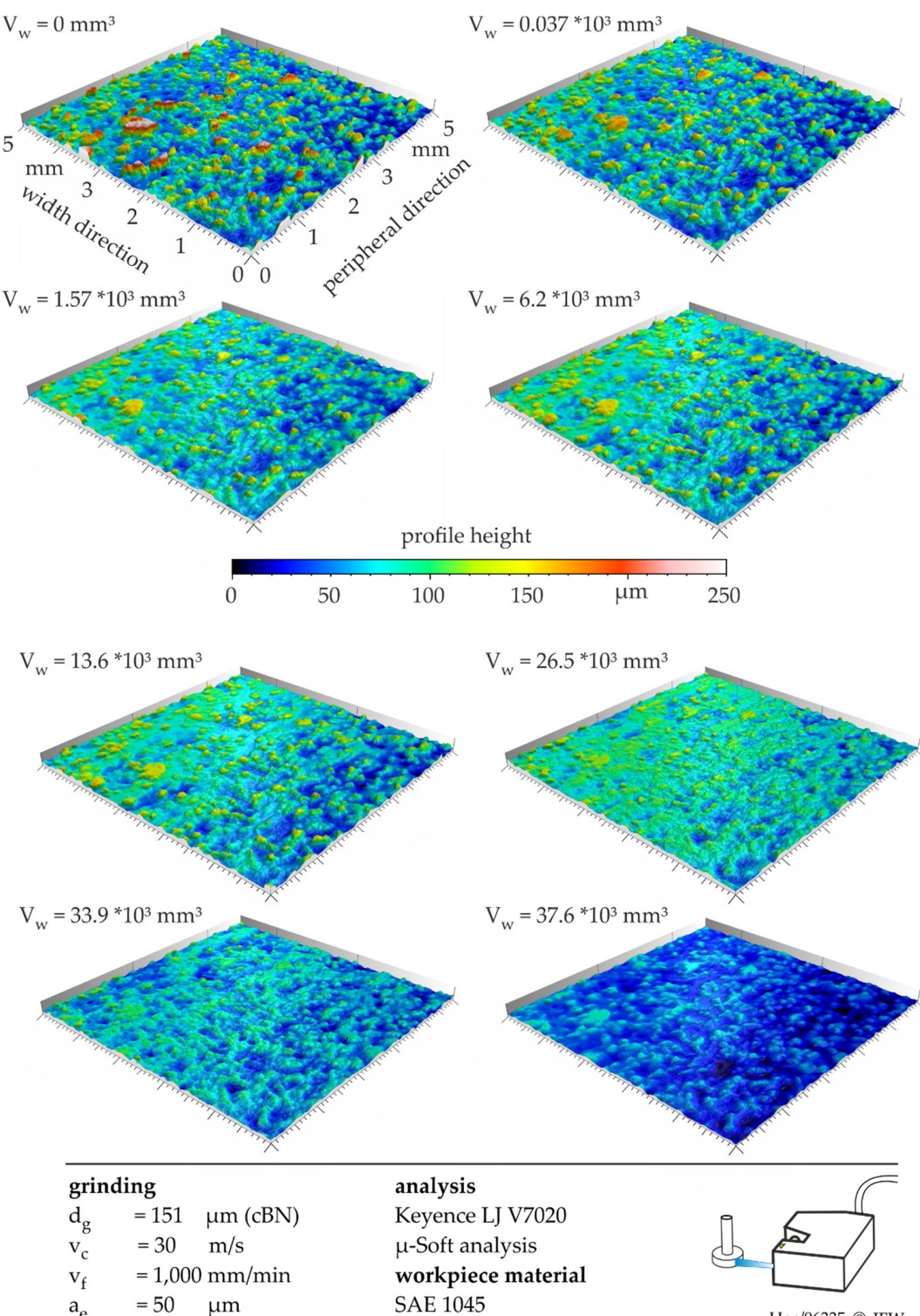

**Figure 4.** Detailed tool topography during wear investigations.

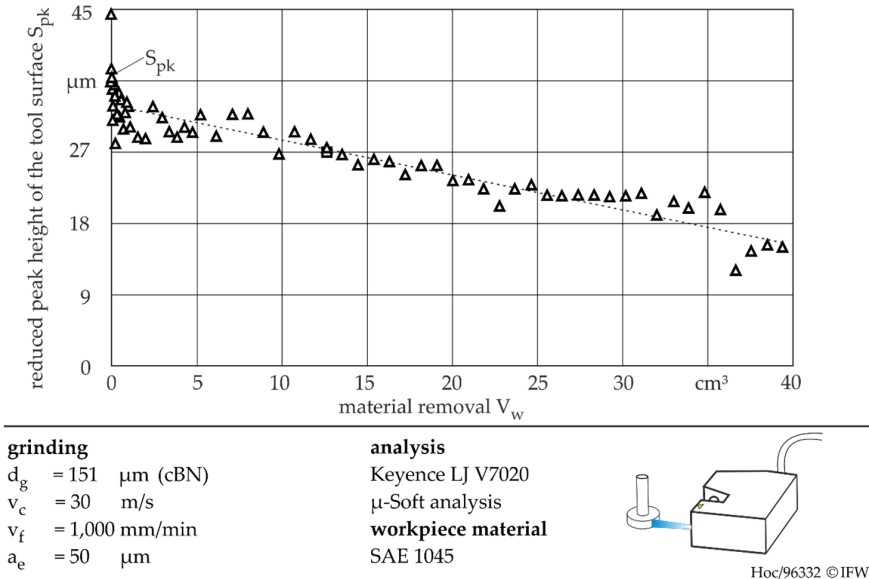

**Figure 5.** Reduced peak height over removed workpiece volume.

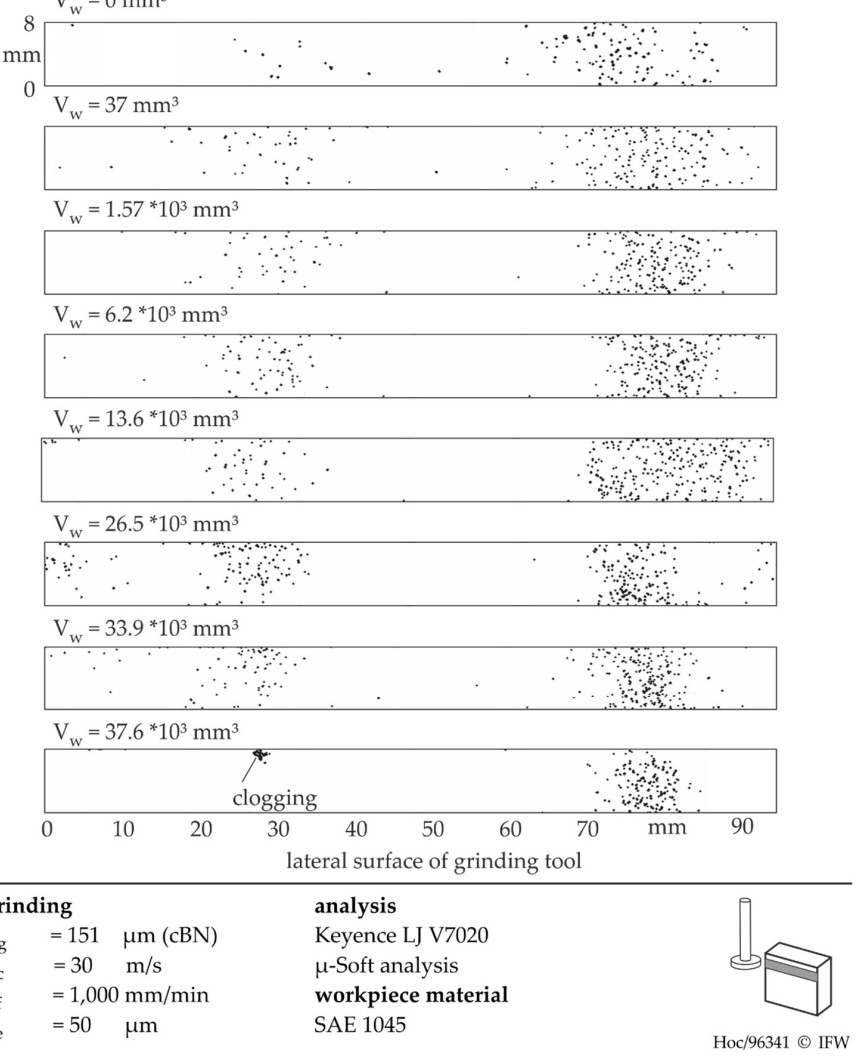

**Figure 6.** Surface-generating tool points.

The present investigation shows the possibilities of grinding tool analysis by measuring the whole tool topography. To prove that the envelope profile theory is correct, the ground workpiece topographies were simulated with varying tool topographies. The simulated workpieces were compared to the related measured workpiece. Figure 7 shows three examples of the 67 comparisons between workpiece and grinding tool. It should be noted that the profile height differs due to the wear state of the grinding wheel. The first comparison shows a correlation of r = 75% and the average surface roughness is predicted with a high accuracy of 98%. Due to the fact that every measuring error from the laser triangulation influences the simulation result, same peaks can be found in the simulated workpiece topography as well. That is why the second comparison in Figure 7 shows only a correlation of 67% and the average surface roughness Rz is overestimated by three times. However, in all three comparisons the characteristic, form and peak position between measured and simulated grinding profile is nearly the same. Due to grain wear and local cloggings, the profiles are not exactly the same in the width direction of the grinding wheel. The graphs show that the envelope profile theory works and that the highest points of the grinding tool topography generate the workpiece surface.

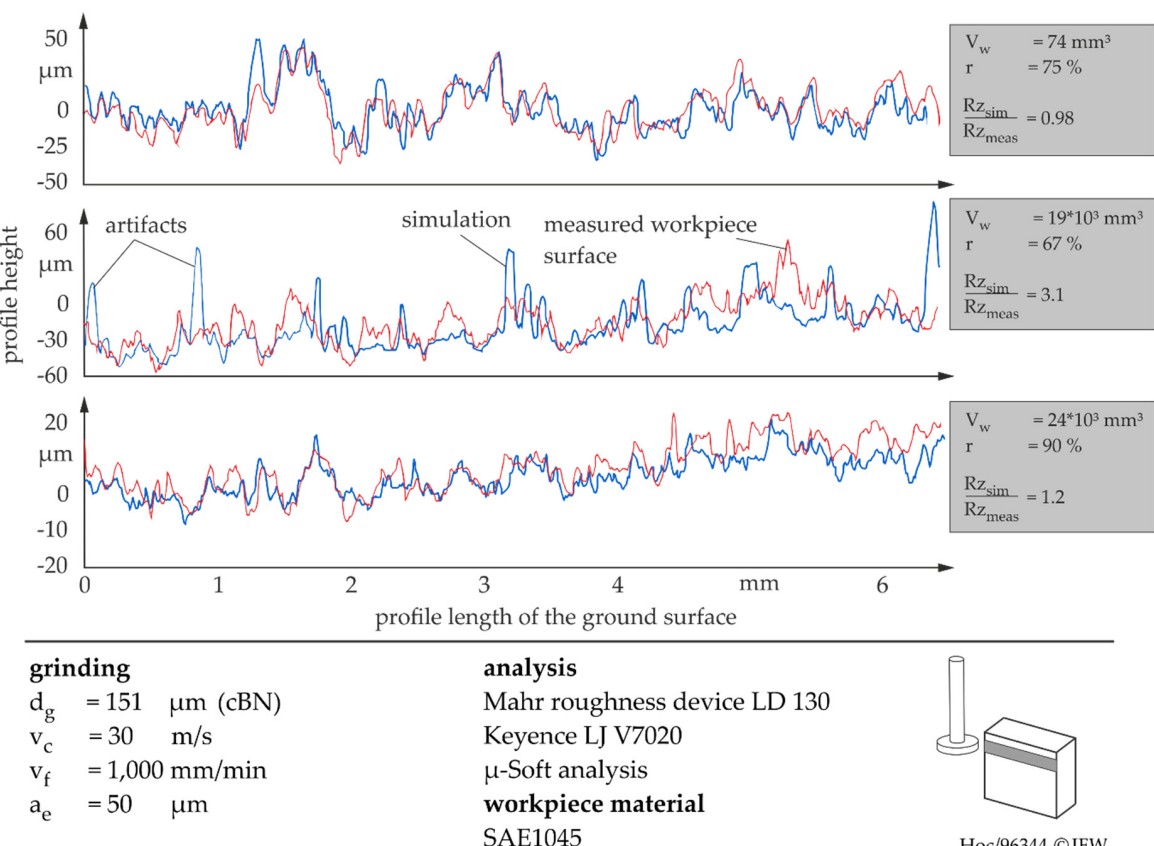

**Figure 7.** Examples of comparison between simulated and measured workpiece topography.

If we take a look at the results of all comparisons (Figure 8), the mean value for the correlation coefficient is about 70%. During the first 1000 mm$^3$ of machined material, the correlation coefficient drops from 83% to less than 60%. This is caused by initial wear and the breakage of the highest grains. After that, the wear behaviour stabilised and the correlation coefficients between the measured and simulated workpiece profile increased again to values of about 80%. Due to the stable height of measuring errors and the decreasing roughness of the tool topography, the average surface roughness values are overestimated with increasing removed workpiece volume. To verify these results and to evaluate the value of the correlation coefficient of 70%, the simulated workpieces are compared to six different additional workpieces, which were ground with different tools with a grain diameter of 151 µm. The red point in Figure 8 shows that the comparison to those workpieces only reaches

values of about 32%. This cross reference confirms the envelope profile theory and the quality of the simulation again. In so far, the presented approach allows for a differentiation of similar grinding processes by investigating the specific resulting workpiece topography.

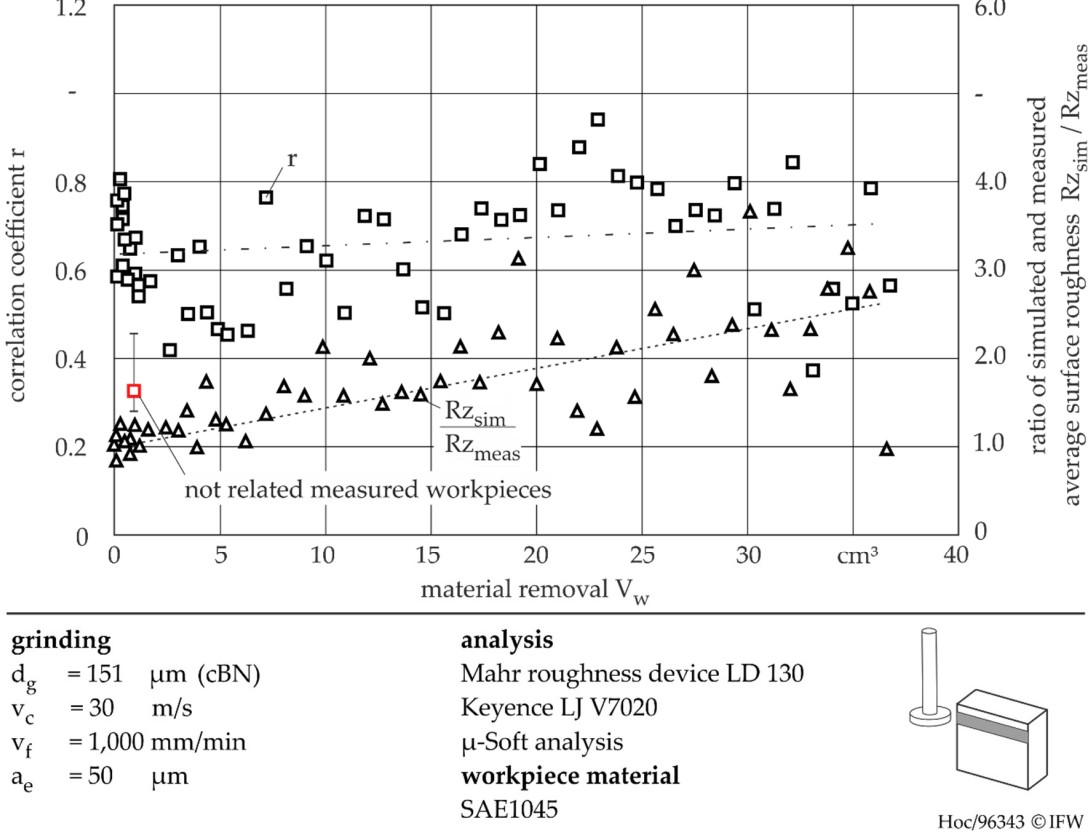

**Figure 8.** Comparison between simulated and measured workpiece topography by means of the average surface roughness Rz and the correlation coefficient r.

## 4. Conclusions

The present approach of measuring the whole grinding tool topography by laser triangulation can describe the wear behaviour with a high accuracy of $10 \times 7.5$ µm (width × circumference). Therefore, the exact position of the surface-generating tool parts can be identified. Especially for galvanically bonded tools, this is important, because of the concentricity error. By using the measured topography in a dexel-based material removal simulation, it was possible to confirm the envelope profile theory for the first time. The mean value for the correlation coefficient between measured and simulated workpiece surfaces is about 70%. The comparison between not related workpieces and the measured grinding wheel reaches only 32%. With this method an assignment of tool and workpiece can be performed, which allows a passive plagiarism for ground components. The correlation between both profiles (grinding wheel and resulting surface) identifies plagiarism. For further applications like plagiarism detection, the measurement accuracy has to be increased and errors, which influences the simulation result, have to be eliminated by more accurate misfit filters.

**Author Contributions:** Conceptualization, Methodology, Validation, Formal Analysis, Investigation, Resources, Data Curation, Writing-Original Draft Preparation, Writing-Review & Editing and Visualization by R.H.; Software by V.B. and R.H.; Supervision by T.G.; Project Administration and Funding Acquisition by B.D.

**Funding:** The German Research Foundation (DFG) within the Collaborative Research Centre (CRC) 653 funded this research. We thank the DFG for its financial support of this project.

**Conflicts of Interest:** The authors declare no conflict of interest.

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
