# Peer review of "Prediction of Ground Surfaces by Using the Actual Tool Topography"

_jmmp, doi:10.3390/jmmp3020040_

Round 1
Reviewer 1 Report
This study used a 3D laser triangulation sensor to obtain a real grinding tool topography in the machining process, and the measured topography was imported into a material removal simulation after a data processing step. A simulation of the real ground surface could be predicted the exact position of the surface-generating tool. Therefore, the roughness profile of grinding could be obtained after the simulation. This is a very interesting research to predict the grinding surface profiles in the precision machining. However, some of the questions in contents need authors to modify or identify in this study.
1. The main results need to be presented in the abstract.
2. Some of grammar errors need to be modified, and some type errors need to be revised, for ex. Line 98, line133 investigated tool (not tool it should abrasive), line 144 what is the 67 comparisions? Is 67 data comparisions?
3. There are 12 references only discuss the tool models in the introduction, however, no grinding phenomena are introducing in the introduction.
4. Authors should clearly define the key words in the manuscript (1) one stroke (2) first iteration (3) what is the meaning of the first time in the end of the abstract?
5. Since the thickness of spraying powders will affect the measuring accuracy of the grinding wheel topography, therefore, what is the particle size of the spray powders? How did authors control the spraying thickness of the core optispray? And how did the software to remove the artifacts and fill the missing area? Authors should state the accuracy after measuring by the software.
6. From this research, no lubricants were used in the grinding process, it seems not very reasonable, why did authors not use lubricants in this study?
Author Response
Dear Reviewer,
thank you for your helpful comments. Following you find my response:
The main results need to be presented in the abstract.
I add the sentense: The correlation between the roughness profiles of the workpiece and envelope profiles of the grinding wheel results in values about 70%.
2. Some of grammar errors need to be modified, and some type errors need to be revised, for ex. Line 98, line133 investigated tool (not tool it should abrasive), line 144 what is the 67 comparisions? Is 67 data comparisions?
Thanks for your comments. I correct it.
3. There are 12 references only discuss the tool models in the introduction, however, no grinding phenomena are introducing in the introduction.
Stuckenholz and Marschalkowski are the first references. They discuss grinding phenomena and the envelope profile theory.
4. Authors should clearly define the key words in the manuscript (1) one stroke (2) first iteration (3) what is the meaning of the first time in the end of the abstract?
Stroke and iteration means one step between measurement and grinding.
5. Since the thickness of spraying powders will affect the measuring accuracy of the grinding wheel topography, therefore, what is the particle size of the spray powders? How did authors control the spraying thickness of the core optispray? And how did the software to remove the artifacts and fill the missing area? Authors should state the accuracy after measuring by the software.
The paper says: In spite of the spray-coating with a particle size of 1 ‑ 5 µm small artifacts occur in the raw data. A misfit filter in µ-Soft analysis was used to remove the artifacts and fill missing areas with interpolated points.
6. From this research, no lubricants were used in the grinding process, it seems not very reasonable, why did authors not use lubricants in this study?
That my mistake. Of course I used lubricant.
I hope the comments answer all your questions.
Thanks a lot.
Best regards
Rolf Hockauf
Reviewer 2 Report
It is a very good job, well written and described. Please just insert one or two figures showing the experimental set up of your machine.
Author Response
Dear reviewer,
thank you for reading my paper. Unfortunately the experimental set up is not that considerable as the set up schema I used.
best regards
Rolf Hockauf
Reviewer 3 Report
Paper title
Prediction of ground surfaces by using the actual tool topography
This work presents a prediction model for ground surfaces that uses the actual grinding wheel topography to carry out the simulation.
The quality of the manuscript is high and can be accepted in the present form
Author Response
Dear reviewer,
thank you for reading my paper and the nice comment.
best regards
Rolf Hockauf
Round 2
Reviewer 1 Report
Question 1. The main results need to be presented in the abstract.
Authors reply: I add the sentense: The correlation between the roughness profiles of the workpiece and envelope profiles of the grinding wheel results in values about 70%.
Question: the above sentence didn’t appear in the abstract.
Question 4. Authors should clearly define the key words in the manuscript (1) one stroke (2) first iteration (3) what is the meaning of the first time in the end of the abstract?
Stroke and iteration means one step between measurement and grinding.
Question: “(3) what is the meaning of the first time in the end of the abstract?” Authors didn’t answer.
6. From this research, no lubricants were used in the grinding process, it seems not very reasonable, why did authors not use lubricants in this study?
That my mistake. Of course I used lubricant.
Question: authors should explain “will the lubricant spraying on the grinding wheel induce the measuring error by laser source?”
Author Response
Question 4. Authors should clearly define the key words in the manuscript (1) one stroke (2) first iteration (3) what is the meaning of the first time in the end of the abstract?
Stroke and iteration means one step between measurement and grinding.
Question: “(3) what is the meaning of the first time in the end of the abstract?” Authors didn’t answer.
For the first time means that the envelope profile theory has never been proven in "history" by using the whole grinding wheel surface.
6. From this research, no lubricants were used in the grinding process, it seems not very reasonable, why did authors not use lubricants in this study?
That my mistake. Of course I used lubricant.
Question: authors should explain “will the lubricant spraying on the grinding wheel induce the measuring error by laser source?”
I think there is a misunderstanding. We used lubricant for the grinding investigations. After that the grinding wheel is dried with compressed air. Now the wheel is spray coated to prevent reflections in the laser measurement. The spray coating has an impact on the errors, but that is part of further investigations.
